# Giant Intraosseous Cyst-Like Lesions of the Metacarpal Bones in Rheumatoid Arthritis

**DOI:** 10.3390/jimaging7070113

**Published:** 2021-07-12

**Authors:** Wanxuan Fang, Ikuma Nakagawa, Kenneth Sutherland, Kazuhide Tanimura, Tamotsu Kamishima

**Affiliations:** 1Faculty of Health Sciences, Hokkaido University, North-12 West-5, Kita-ku, Sapporo 060-0812, Japan; fangwanxuan2020@163.com; 2Department of Rheumatology, Hokkaido Medical Center for Rheumatic Diseases, Kotoni 1-3, Nishi-ku, Sapporo 063-0811, Japan; ikuma.nakagawa@gmail.com (I.N.); k.tanimura@pep.ne.jp (K.T.); 3Global Center for Biomedical Science and Engineering, Hokkaido University, North 15 West 7, Kita-ku, Sapporo 060-8638, Japan; kensuth@med.hokudai.ac.jp

**Keywords:** rheumatoid arthritis, magnetic resonance imaging, intraosseous cyst, metacarpal bone, bone erosion

## Abstract

The purpose of this study was to illustrate the clinical and imaging properties of giant intraosseous cyst-like lesions (GICLs) of the metacarpal bones extending beyond the central diaphysis in rheumatoid arthritis (RA) patients on magnetic resonance (MR) images. A keyword search was conducted to extract GICLs of the metacarpal bones out of MR reports in RA patients. There were nine GICLs extending from the subchondral bone region beyond the central diaphysis of the metacarpal bones on MR images in eight subjects with RA (seven females, one male). The age range was from 60 to 87 years with a median age of 65.5 years. The average disease duration was 13.1 years. As for the disease activity, one was low, six were moderate and one was high. None of the nine lesions were visible on radiography. The Steinbrocker stage distribution was as follows: I (*n* = 3), II (*n* = 2), and III (*n* = 3). Intraosseous cyst-like lesion of the metacarpal bones on MR images is a relatively rare manifestation in patients with long-standing RA. Although the lesion seems to be derived from subcortical bone break, it is not necessarily erosive in nature.

## 1. Introduction

Rheumatoid arthritis (RA) is a chronic inflammatory systemic autoimmune disease that compromises the joints, causing cartilage destruction and bone damage [1]. This disease affects approximately 1% of the population and occurs in women two to three times more frequently than in men [2]. Although it can occur at any age, the peak incidence is at the age of 50 years [3].

Bone erosion as well as joint space narrowing is the main imaging manifestation of bone and joint damage in RA [4]. Synovitis, tenosynovitis [5], and bone marrow edema [6] are common inflammatory lesions of RA. Patients with long-standing, inadequately treated RA develop joint damage and deformities, including characteristic ulnar deviation, swan neck, and boutonniere deformities of the finger joints, and flexion contractures of the knees and elbows [7].

Synovial cysts represent an abnormal distension of bursae [8], which may or may not communicate with the adjacent joint [9]. Histology shows that the walls of subchondral cyst-like lesions are composed of loose fibrous and connective tissue with no epithelial or synovial cell lining over its inner aspect. In some places, small vessels with thickened walls are seen, as well as a poorly organized polymorphic cellular infiltrate. Necrotic material and inflammatory tissue similar to the pannus are found within the lesion [10]. The term “intraosseous synovial cyst” is used to designate the epiphyseal cyst-like lesions seen in a variety of clinical settings [11]. Intraosseous synovial cysts that develop from rheumatoid arthritis (RA) are a rare condition [12].

In this report, we present and discuss giant intraosseous cyst-like lesions (GICLs) of the metacarpal bones extending from the joint surface beyond the central diaphysis on magnetic resonance (MR) images in light of radiographic findings and clinical manifestations. In spite of the lack of pathological proof due to their indolent nature, these lesions are presumed to be intraosseous synovial cysts when taking imaging findings and clinical presentations into consideration. We also touch on imaging differentials of this rare manifestation of RA.

## 2. Materials and Methods

Ethical permission was obtained at a hospital specialized in rheumatology. Informed consent was obtained in the form of an opt-out on the website of the hospital. In this retrospective study, we searched for more than 1000 MR imaging examinations of the hand in the local hospital from January 2012 to April 2021. Eight subjects (seven females, one male) with RA were identified by a keyword search of “large erosion”, “large cyst”, “metacarpal bone”, and “rheumatoid arthritis” out of radiological reports in that period. The age range was from 60 to 87 years old, and the median age was 65.5 years old. The selection criteria are patients who had GICL of the diaphysis and fulfilled the 2010 American College of Rheumatology and the European League Against Rheumatism classification criteria for RA [13]. GICL of the diaphysis was defined as a well-demarcated cystic lesion extending from the joint surface, usually with cortical break beyond the central diaphysis of the metacarpal bones.

We reviewed clinical symptoms, duration of disease and laboratory examinations in eight subjects of GICL. The laboratory data included rheumatoid factor (RF), anti-cyclic citrullinated peptide antibody (ACPA), C-reactive protein (CRP), the levels of the erythrocyte sedimentation rate (ESR), matrix metalloproteinase-3 (MMP-3), and disease activity score (DAS), assessed as DAS28-ESR and DAS28-CRP.

Three of the subjects suffered from comorbidities: pustulotic arthro-osteitis (*n* = 1), Sjogren’s syndrome (*n* = 1), and Still’s disease (*n* = 1). All subjects were treated with nonsteroidal anti-inflammatory drugs (NSAIDs), disease-modifying antirheumatic drugs (DMARDs), and/or glucocorticoid before MR imaging (MRI) examination.

Unilateral hands were scanned on a 0.3T MR system (AIRIS Vento, Hitachi, Tokyo, Japan) using a knee coil, while their corresponding hand radiography images were compared (*n* = 8). The hand radiography was scored by using the Steinbrocker stage [14]. All radiography and MR images were examined with consensus by a radiologist who has specialized in rheumatology imaging for more than 20 years and a rheumatologist with 10 years of experience. Corresponding joint ultrasound records were also reviewed.

Spin echo T1 weighted coronal images (repetition time/echo time, 400/28 ms; field of view, 250 mm; matrix, 256 × 134; 14 slices; slice thickness, 3 mm) and short tau inversion recovery (STIR) coronal images (repetition time/echo time/inversion time, 4000/22/110 ms; field of view, 250 mm; matrix, 256 × 114; slice thickness, 3 mm) were obtained for the dominant hand. In addition, two subjects underwent a contrast-enhanced fat-suppressed T1-weighted spin-echo sequence (repetition time/echo time, 500/21.4 ms; field of view, 250 mm; matrix, 512 × 512; slice thickness, 3 mm) for hand examination. With regard to the contrast agent, before 2018 we used gadodiamide (Omniscan^®^; Daiichi Sankyo Co. Ltd., Tokyo, Japan) and after 2018 we used gadoteridol (ProHance^®^, Bracco Eisai Co. Ltd., Tokyo, Japan) at a standard dosage.

An ultrasonographic system (EUP-L34P, HI VISION Avius; Hitachi, Tokyo, Japan) equipped with a 13-MHz linear array transducer was used for hand joint examination. The 1st to 5th metacarpophalangeal (MP) and 1st to 5th proximal interphalangeal (PIP) joints were scanned in the longitudinal plane over the dorsal surface. Semiquantitative scoring for power Doppler mode was used to assess joint synovitis (grade 0, absence of signal; grade 1, single vessel dots; grade 2, vessel dots over less than half of the synovium; and grade 3, vessel dots over more than half of the synovium) [15].

## 3. Results

The basic characteristics of the research population, such as age, sex, radiological features, clinical manifestations, duration of disease, and the Steinbrocker stage, are shown in Table 1. The laboratory examinations are depicted in Table 2.

In this study, we found nine GICLs of the metacarpal bones extending beyond the central diaphysis on MR images in eight subjects. The GICLs originated from the proximal (carpometacarpal) joint surface (being involved in five subjects: Figure 1, Figure 2 and Figure 3) and from the distal metacarpophalangeal joint surface in two subjects (Figure 4 and Figure 5). The GICLs were in the first (Figure 1, Figure 3 and Figure 5), the second (Figure 2), and the third (Figure 2 and Figure 4) metacarpal bones. GICLs were found in both the second and third metacarpal bones in one subject. For the rest of the subjects, GICLs were detected in a single metacarpal bone. One of the patients (Figure 1) had local osteoarthritis (OA) related with the GICL. 

On radiography (Figure 1c, Figure 3d, Figure 4d and Figure 5e), we noticed no thinning of the bone cortex or remodeling at the site of the GICL. Joint space narrowing (JSN) was seen in three subjects. One subject presented with carpal collapse and about a 1-cm deformity or dislocation. The remaining had no radiographic abnormalities. In addition, none of the GICLs were visible on radiography. Based on the Steinbrocker stage, three subjects (Figure 3 and Figure 4) were classified as stage I, two (Figure 5) subjects were classified as stage II, and three subjects (Figure 1 and Figure 2) were classified as stage III. 

On MR imaging, we found that GICL had clear borders and nonspecific internal characteristics without septum-like structures, clots, nor debris, and detected no bone marrow edema-like signal intensity around the GICL.

The joint ultrasound records revealed that there were no subjects with blood flow signals in MP joints adducent to the cystic bone lesion. No information on carpometacarpal joints was available because they were not included in the ultrasonographic routine practice. None of the subjects had a history of local trauma. Additionally, no subject had local symptoms such as pain and swelling when examined for the first time. Four of them still had no corresponding pain or swelling in the follow-up examination, and the remaining half suffered from varying degrees of pain or swelling.

The DAS28-ESR was available in all subjects. Based on the EULAR response criteria, the values of DAS28 < 2.6, DAS28 ≤ 3.2, DAS28 ≤ 5.1, and DAS28 > 5.1 are defined as remission, low, moderate, and high disease activity state [16], respectively. One subject had a low (Figure 1), six subjects (Figure 2, Figure 3, Figure 4 and Figure 5) had a moderate, and one subject had a high disease activity.

## 4. Discussion

In this study, we reviewed nine GICLs of the metacarpal bones extending beyond the central diaphysis on MR images in eight subjects, none of which were detected on radiography. The lesions appeared to originate from the adjacent joint and were mostly found in middle-aged and elderly women without any episode of trauma. The disease duration and extent of destructive changes in the affected RA patients were variable, and the majority of them had a moderate disease activity. 

To the best of our knowledge, identical lesions in the diaphysis of the metacarpal bone related with RA have not been reported in the literature; however, similar imaging features in the other bones have been reported. Amin et al. [11] reported a 56-year-old woman with RA affected with a giant intraosseous cystic lesion in the distal tibia communicating with the ankle joint on MR images (neither pathological proof nor radiography was available). Lohse et al. [10] reported a 53-year-old woman with RA who had an intraosseous cyst in the epiphyseal and metaphyseal part of the left tibia, which was a sharply marginated oval lesion with no sclerotic rim and no clear evidence of an overlying periosteal reaction on radiography.

We are more specific in the imaging findings of GICL of metacarpal bones in RA patients. In our study, we could see low signals which extended to the central diaphysis of the metacarpal bone on T1-weighted coronal images. Additionally, on STIR coronal images, high signals which had clear boundaries could be seen. Contrast-enhanced fat-suppressed T1-weighted images can differentiate fluid-filled cysts from pannus-filled spaces. The latter are highly vascularized and enhance greatly on contrast-enhanced fat-suppressed T1-weighted images, whereas a “cyst” is by definition a fluid-containing structure with a homogeneous low signal on T1-weighted images and a homogeneous high signal on T2-weighted images. The cyst may enhance only at peripheral margins if images are acquired immediately after contrast injection [17]. Contrast-enhanced fat-suppressed T1-weighted images showed an enhancement of the peripheral margins of the lesion compatible with a cyst in two subjects. 

At present, there are two hypotheses to explain the genesis of these cysts. One is explained by joints which have raised intra-articular pressure within the joint exceeding that of adjacent intraosseous pressure. The concurrent loss of articular cartilage results in the development of defects in the articular cartilage resulting in the migration of synovial fluid into the underlying subchondral bone [18]. The other hypothesis is intraosseous rheumatoid nodules [17]. In our study, we are leaning toward the former hypothesis as no subjects had a blood flow signal in the local diseased area in the ultrasonographic examinations.

In our series, GICLs of metacarpal bones were not seen on radiography. The reason is probably because radiography is more suitable for detecting diseases with cortical remodeling or destruction. By contrast, the cortical erosions were too minor to be detected in our cases, or the cortical erosions had been healed. The absence of cortical destruction is consistent with the presumed origin of the hypothesis that synovial fluid resulted in the rise of intramedullary pressure. Experiments [19,20] have proved that the increase of intramedullary pressure elicited new bone formation. The new bone repaired the margin of the eroded cortical bone. Therefore, cortical destruction of metacarpal bones cannot be seen. 

From the records of clinical manifestations, we found that no subject had a history of local swelling or pain at the local lesion in their initial episodes. Half of the subjects had intermittent or transient pain or swelling in the years after the onset of giant intraosseous cyst-like lesions. Combined with the subjects’ clinical data and laboratory results, the severity of GICL had no significant relationship with clinical symptoms and disease activity. This suggests that GICL does not show local invasive behaviors and that the overall clinical manifestations are mild. There is no evidence that the severity of the local lesion is related to the duration of disease. In fact, there were two subjects in the follow-up for whom the GICL decreased in size. 

These GICLs are rarely seen in RA patients, and they can easily be confused with trauma and other cystic lesions. It is therefore necessary to differentiate them from other diseases in order to make an accurate diagnosis (Table 3).

Our study has several limitations. The first and major one is that a pathological examination of the lesion was not carried out due to the limitation of the clinical practice for which this was not ethically allowed. To address this matter, we combined the clinical manifestations and imaging findings to characterize the disease. The second limitation is the absence of a functional performance of hands. Although most subjects had no or mild local symptoms such as pain and swelling, we could not assess the impact of GICL on their hand functions. The third limitation is that we only used the 0.3T MR system with a knee coil for examination. More precise observations might be possible if the MR examinations were carried out under MR systems with a higher magnetic field strength.

In conclusion, our study reported the imaging property of GICL of the metacarpal bone extending beyond the central diaphysis on MR images in RA patients and its differential diagnosis. Since GICL is easily misdiagnosed as trauma or cystic lesions, especially when MR imaging is performed without a corresponding radiography, being aware of this lesion in RA can be clinically significant and crucial for every radiologist and rheumatologist.

## Figures and Tables

**Figure 1 jimaging-07-00113-f001:**
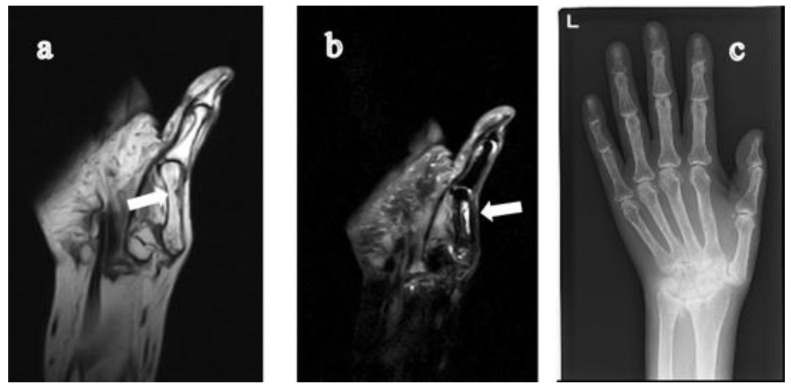
60-year-old woman with rheumatoid arthritis for 28 years. There is a cystic lesion in the 1st metacarpal bone of the left hand, which originates from the head of the 1st metacarpal bone extending to the central diaphysis. The lesion appears as a low signal intensity on the T1-weighted coronal image (**a**) and high signal intensity on the STIR coronal image (**b**). On CR (**c**), collapse of the carpal bone due to long-standing RA is observed; however, the lesion is not visible. RA, rheumatoid arthritis; STIR, short tau inversion recovery; CR, computed radiography.

**Figure 2 jimaging-07-00113-f002:**
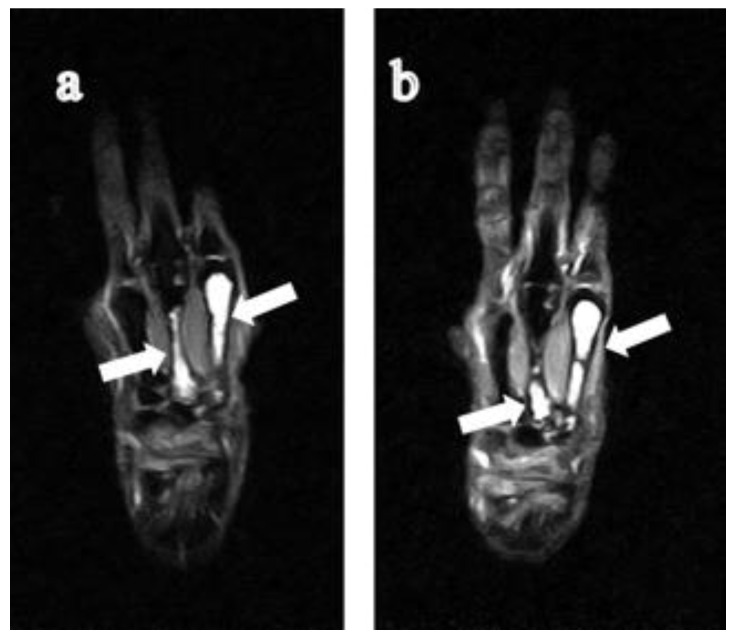
64-year-old woman with rheumatoid arthritis for five years. (**a**) STIR coronal image shows that the cystic lesions originated from the proximal of the 2nd and 3rd metacarpal bones of the left hand respectively and extended to the central diaphysis. (**b**) During follow-up, the giant intraosseous synovial cyst of the middle finger became smaller. RA, rheumatoid arthritis; STIR, short tau inversion recovery.

**Figure 3 jimaging-07-00113-f003:**
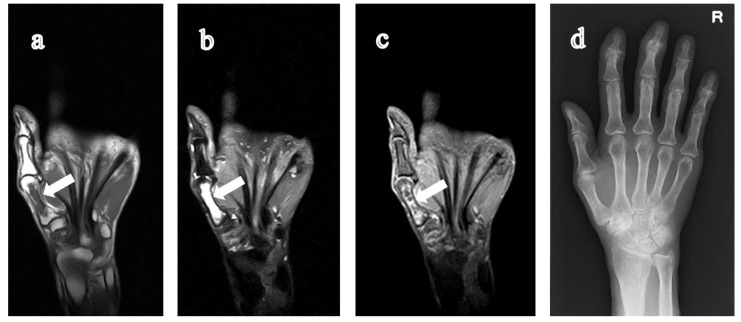
84-year-old woman with rheumatoid arthritis for six years. On T1-weighted coronal image (**a**) and STIR coronal image (**b**), a giant lesion is detected, which originated from the proximal aspect of the 1st metacarpal bone of the right hand and extended to the central diaphysis. (**c**) The internal enhancement of the cystic lesion can be seen on the contrast-enhanced fat-suppressed T1-weighted coronal image. (**d**) The lesion cannot be detected on CR. RA, rheumatoid arthritis; STIR, short tau inversion recovery; CR, computed radiography.

**Figure 4 jimaging-07-00113-f004:**
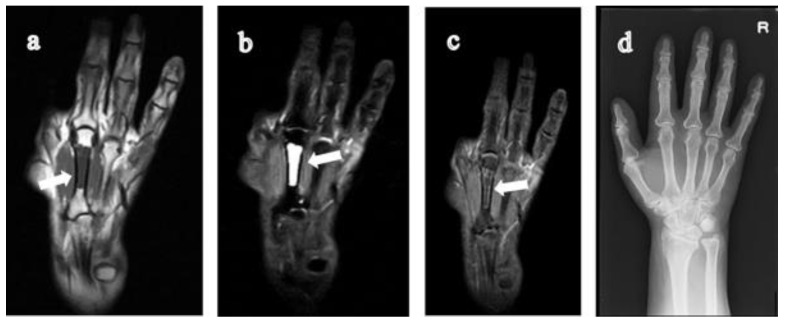
62-year-old woman with rheumatoid arthritis for one year. (**a**) T1-weighted coronal image of the 3rd metacarpal bone of the right hand shows a low signal lesion (solid arrow) that originated from the distal and extended to the central diaphysis of the metacarpal bone. On STIR coronal image (**b**), the lesion has a high signal intensity (solid arrow) with clear boundaries. (**c**) Contrast-enhanced fat-suppressed T1-weighted coronal image shows an enhancement of peripheral margins. On CR (**d**), the bone lesion is not visible. RA, rheumatoid arthritis; STIR, short tau inversion recovery; CR, computed radiography.

**Figure 5 jimaging-07-00113-f005:**
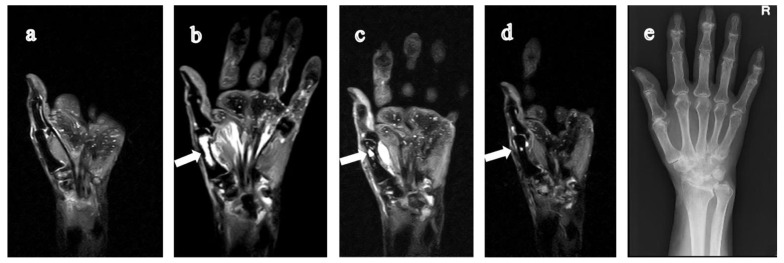
67-year-old woman with rheumatoid arthritis for eight years. At the baseline (**a**), there was no abnormality in the 1st metacarpal bone of the right hand on coronal STIR. Nine months later, (**b**) a high signal lesion appeared. The lesion originated from the distal aspect of the 1st metacarpal bone of the right hand and extended to the central diaphysis with synovitis, tenosynovitis, and extraosseous cysts. After another six months, (**c**) the cystic lesions in the 1st metacarpal bone dramatically decreased in size. (**d**) The cystic lesions remained stable during the follow-up of six years. (**e**) The cyst lesion cannot be seen on CR throughout this period. RA, rheumatoid arthritis; STIR, short tau inversion recovery; CR, computed radiography.

**Table 1 jimaging-07-00113-t001:** Basic clinical characteristics of subjects.

Subject	1	2	3	4	5	6	7	8
Age (yr)	60	62	67	64	64	84	87	84
Sex	F	F	F	F	F	M	F	F
Laterality	l	r	r	r	r	l	r	l
Location (meta-carpal bone)	1st	3rd	1st	1st	2nd	1st	3rd	2nd and 3rd
Origin	proximal	distal	distal	proximal	proximal	NA	proximal	proximal
OA changes	y	n	n	n	n	n	n	n
Radiography	not visible	not visible	not visible	not visible	not visible	not visible	not visible	not visible
Steinbrocker stage	3	1	2	1	3	1	2	3
Disease Duration (yr)	34	1	8	5	28	4	19	6
History of local trauma	nil	nil	nil	nil	nil	nil	nil	nil
Local symptoms	nil	nil	nil	nil	nil	nil	nil	nil
Complicated with other disease	nil	PAO	SJS and Still’s disease	nil	nil	nil	nil	nil

yr: year; F: female; M: male; l: left; r: right; y: yes; n: no; NA: not available; OA: osteoarthritis; Local symptom: pain and swelling; PAO: pustulotic arthro-osteitis; SJS: Sjogren’s syndrome.

**Table 2 jimaging-07-00113-t002:** Laboratory examination.

Subject	1	2	3	4	5	6	7	8
RF (cut-off 15 IU/mL)	7	263	0	0	232	34	122	25
ACPA (cut-off 4.5 U/mL)	10	4.8	0	0	>1200	1130	199	3.7
CRP (mg/dL)	0.05	0.4	4.04	0.03	1.47	3.38	9.34	0.74
ESR (mm/h)	47	35	49	17	18	92	85	39
MMP-3 (ng/mL)	53.8	42	288	16.7	132	542	209	93.9
DAS28-CRP	1.11	3.10	3.63	2.61	3.56	4.89	4.29	2.51
DAS28-ESR	2.70	3.96	4.05	3.60	3.64	5.81	4.80	3.35

RF: rheumatoid factor; ACPA: anti-cyclic citrullinated peptide antibody; CRP: C-reactive protein; ESR: erythrocyte sedimentation rate; MMP-3: matrix metalloproteinase-3; DAS28: disease activity score in 28 joints scoring in rheumatoid arthritis.

**Table 3 jimaging-07-00113-t003:** Differential diagnosis.

Type of Disease	Age	Common Site	Radiography Characteristics	MR Imaging Characteristics
Bone bruise caused by trauma	Any age	Any site	cannot be seen	low signal intensity on T1-weighted and hyperintensity on STIR images [21]
Enchondroma	20–50 y	centrally in the phalanges of hands and feet, femur, humerus and metacarpals	local osteolytic lesions with calcification [22]	lobulated contours [23] with low signal intensity on T1-WI and very high signal intensity on T2-WI with fat suppression
Brodie Abscess	child	metaphysis of tubular bones	well-defined round or oval cavity with a thin rim of sclerosis [24]	penumbra sign [25] with low signal intensity on T1-weighted and high on T2-weighted and STIR images
ABC	all ages, preferably < 20 y	central or eccentric origin in the metaphysis or diaphysis of a long bone	a fusiform expansile osteolytic lesion with thinned-out cortex [26]	fluid-fluid level [26] on T1-weighted image and high signal intensity on T2-weighted images
FD	10–30 y	femur, tibia, humerus, skull, ribs	greyish ground-glass appearance and usually bound by a characteristic thick layer of reactive bone that has been described as a ‘‘rind’’ [27]	sharply demarcated borders and intermediate to low signal intensity on T1-WI and intermediate to high intensity on T2-WI
Giant Cell Tumors of Bone	20–40 y	Knee: femur, patella, tibia, and fibulaHand: metacarpals	meta/epiphyseal osteolytic lesions that extend to the subarticular cortex of the bone [28]	low signal intensity on T1-WI and high signal intensity on T2-WI
NTM	any age	lung, 5–10% musculoskeletal system	preferential involvement of the metaphyses and diaphyses of long bones, multiple sites of involvement, and discrete lytic areas with marginal sclerosis and osteoporosis [29]	low signal intensity on T1-WI and high signal intensity on T2-WI and STIR sequences

ABC: aneurysmal bone cyst; FD: fibrous dysplasia; OS: osteosarcoma; NTM: nontuberculous mycobacterial; y: years; STIR: short inversion time inversion-recovery.

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
