# Peer review of "Giant Intraosseous Cyst-Like Lesions of the Metacarpal Bones in Rheumatoid Arthritis"

_2313-433X, 2021, doi:10.3390/jimaging7070113_

Round 1

Reviewer 1 Report

The authors describe in particular detail the radiological aspects of a rare bone alteration found in patients with rheumatoid arthritis. The description of the mri data, the pathogenetic hypotheses, the characteristics of the enrolled patients are very precise.

Author Response

Comments from Reviewer 1

The authors describe in particular detail the radiological aspects of a rare bone alteration found in patients with rheumatoid arthritis. The description of the mri data, the pathogenetic hypotheses, the characteristics of the enrolled patients are very precise.

Thank you so much for an encouraging comment.

As Reviewer 1 has suggested that English language and style of our manuscript are fine/minor spell check required, we addressed this issue during revision.

Reviewer 2 Report

 This manuscript describes the observation of a GIOC in the metacarpal bones of a patient with RA. The GIOCs are not visible on plain radiographs, but are visible on MRI. The GIOC does not seem to be related to clinical symptoms or disease activity. The authors disclose the cystic change has no blood flow inside. As the authors stated, the lack of pathological proof is weak point.

Abstract

No comment

Introduction

No comment

Materials and methods

1. How and by whom were the plain radiographs evaluated?

2. Were there any changes on the plain radiographs at all? Thinning of the bone cortex, etc.

3. No evaluation of GIOC morphology (borders, etc.) and internal characteristics (e.g., septum-like structures, clots, debris, etc.). No bone marrow edema-like signal intensity around the GIOCs?

Discussion

The mechanism of formation of GIOC is speculated to be related to intraarticular pressure, but the relationship between GIOC and joints is not described. It would be better to elaborate on the mechanism and speculation of GIOC formation.

References

No comment

Figures

Good  

Author Response

Comments from Reviewer 2

This manuscript describes the observation of a GIOC in the metacarpal bones of a patient with RA. The GIOCs are not visible on plain radiographs, but are visible on MRI. The GIOC does not seem to be related to clinical symptoms or disease activity. The authors disclose the cystic change has no blood flow inside. As the authors stated, the lack of pathological proof is weak point.

Reply:

Thank you so much for your comment.

Abstract

No comment

Introduction

No comment

Materials and methods

  1. How and by whom were the plain radiographs evaluated?

Reply:

Thank you for pointing this out. All radiography and MR images were examined in consensus by a radiologist who has specialized in rheumatology imaging for more than 20 years and a rheumatologist with 10 years of experience.

  1. Were there any changes on the plain radiographs at all? Thinning of the bone cortex, etc.

Reply:

Thank you for pointing this out. On radiography, we noticed no thinning of the bone cortex or remodeling at the site of the GICL. This was added to the manuscript.

  1. No evaluation of GIOC morphology (borders, etc.) and internal characteristics (e.g., septum-like structures, clots, debris, etc.). No bone marrow edema-like signal intensity around the GIOCs?

Reply:

Thank you for your suggestion. On MRI, we found GIOC has clear borders and non-specific internal characteristics without septum-like structures, clots, and debris, and detected no bone marrow edema-like signal intensity around the GIOC. This was added to the manuscript.

Discussion

The mechanism of formation of GIOC is speculated to be related to intraarticular pressure, but the relationship between GIOC and joints is not described. It would be better to elaborate on the mechanism and speculation of GIOC formation.

Reply:

Our speculation regarding the mechanism of GICL formation is shift of joint fluid due to increased intraarticular pressure into the diaphysis of the metacarpal bone through the subchondral bone defect. Please find this in the 4th paragraph of discussion section: At present there are two hypotheses to explain the genesis these cysts. One is explained by joints which have raised intra-articular pressure within the joint exceeding that of adjacent intraosseous pressure. The concurrent loss of articular cartilage results in the development of defects in the articular cartilage resulting in migration of synovial fluid into the underlying subchondral bone [18]. The other is intraosseous rheumatoid nodules [17]. In our study, we are leaning toward the former hypothesis as no subjects had blood flow signal in the local diseased area in the ultrasonographic examinations.

References

No comment

Figures

Good